# Development of a Systematic qPCR Array for Screening GM Soybeans

**DOI:** 10.3390/foods10030610

**Published:** 2021-03-13

**Authors:** Saet-Byul Park, Ji-Yeong Kim, Do-Geun Lee, Jae-Hwan Kim, Min-Ki Shin, Hae-Yeong Kim

**Affiliations:** 1Institute of Life Sciences and Resources and Graduate School of Biotechnology, Kyung Hee University, Yongin 17104, Korea; 791129@naver.com (S.-B.P.); ndisl3@gmail.com (J.-Y.K.); dob9395@naver.com (D.-G.L.); 2Pinucle, Yongin 16870, Korea; jhkim0105@nate.com; 3Novel Food Division, National Institute of Food and Drug Safety Evaluation, Cheongju 28159, Korea; mkhj1212@korea.kr

**Keywords:** GMOs, screening, qPCR, prediction system, monitoring

## Abstract

A screening method using the 35S promoter and nos terminator for genetically modified organisms (GMOs) is not sufficient to cover all GM soybean events. In this study, a real-time polymerase chain reaction (also known as quantitative polymerase chain reaction, qPCR) array targeting eight screening assays combined with a prediction system was developed for the rapid tracking of GM soybeans. Each assay’s specificity was tested and confirmed using 17 GM soybean events that have been approved in Korea. The sensitivity of each assay was determined to range from 0.01% to 0.05% using DNA mixtures with different GM ratios, and it was validated by the results of three experimenters. The applicability of this study was tested by monitoring 23 processed foods containing soybeans. It was figured out that 13 of the 23 samples included GM soybeans. The prediction system combined with screening results will be helpful to trace the absence/presence of GM soybean events. This new qPCR array and prediction system for GM soybean detection provides rapid, convenient and reliable results to users.

## 1. Introduction

Soybeans (*Glycine max*) are a popular food material in Korea and much of Asia because of their high protein content. Soybeans are consumed as a variety of food products, including tofu, soy sauce, oil, soy protein, doenjang (soybean paste), noodles and soy milk. Korea is highly dependent on soybean imports to supply the market. According to the Korea Biosafety Clearing House, one million tons of GM soybeans were imported in 2019, accounting for half of the GM crops imported into Korea. Therefore, the government and relative institution must have the best possible screening methods for border quarantine and monitoring of GM soybeans. 

The screening method used in Korea tests for the following three targets: the endogenous gene, the Cauliflower Mosaic Virus 35S promoter (P-35S), and the *Agrobacterium tumefaciens* nopaline terminator (T-nos). However, the screening method can detect only five GM soybean events that contain P-35S or T-nos; there are 12 other GM soybean events that should be tested for using event-specific methods at the screening step. This is because the coverage of the screening method is lowered due to the diversification of the genetic elements used for the development of GM soybeans. For this reason, researchers have been working to develop effective screening methods with high coverage.

DNA-based detection methods (such as polymerase chain reaction (PCR)) are the standard method for GMO analysis because of their accuracy and high sensitivity [1,2,3,4]. Various PCR methods (such as multi-plex, loop-mediated isothermal amplification (LAMP), real-time and digital) have been applied to the screening of GMOs. Multi-plex PCR is becoming a popular method, due to its high coverage and simplicity. Randhawa et al. [5] reported a multi-plex screening method targeting selectable marker and reporter genes. Park et al. [4,6] developed crop-specific screening methods for GM soybeans and maize using multi-plex PCR. Singh et al. [7] developed a cost-efficient screening strategy using a crop-specific GMO matrix for GM maize and cotton events.

The real-time PCR assay is also the most useful tool for qualifying and quantifying GMOs. Many researchers have reported real-time PCR screening methods with new screening targets [8,9,10,11,12]. To save time and cost, a pre-spotted plate (PSP) (defined as ready-to-use real-time PCR plates whose wells have been spotted with primers and probes targeting a sequence of interests [13]) has been developed. Querci et al. [14] reported the real-time PCR-based ready-to-use multi-target analytical system to detect several GM events in a single experiment. They described a ready-to-use system using PSP containing lyophilized primers and probes that reduce laboratory handling steps to a minimum. The same group tried to develop a semi-quantification assay for GM maize using a ready-to-use RTi–PCR plate which allows the qualitative detection of GM event in one single real-time PCR experiment [15]. For the detection of unapproved GMOs in Japan, Mano et al. [16] suggested a real-time PCR array as a universal platform. Many researchers also provided a decision-supporting system (DSS) which is a database concerning the genetic elements of GMOs [17,18,19,20,21].

To develop a systemic qPCR array for the screening of GM soybeans, we selected a ready-to-use system combined with a decision support tool, as already reported in various countries [14,16,22]. Since this study only targets soybeans, unlike other studies, the screening assays were limited to eight for convenience and efficiency. The developed qPCR array’s applicability was confirmed by analyzing 23 processed foods containing soybeans as an ingredient. Using this screening qPCR array, the presence or absence of GM soybeans in a sample can be easily determined without any specific knowledge of GMOs.

## 2. Materials and Methods

### 2.1. Preparation of Samples

Seventeen GM soybean events (RRS, DP356043-5, DP305423-1, DAS-68416-4, DAS-44406-6, DAS-81419-2, MON87705, MON87708, MON87701, MON87769, MON89788, CV127, FG72, A2704-12, A5547-127, SYHT0H2 and MON87751) were selected as negative and positive controls for specificity testing. Certified reference materials (CRMs) of GM soybeans were purchased from the Institute for Reference Materials and Measurements (IRMM, Geel, Belgium) and the American Oil Chemists’ Society (AOCS, Urbana, IL, USA).

Twenty-three processed food samples (such as grain powder, tofu, soy milk and snacks) were purchased from Korea (20 samples), Japan (2 samples), and Thailand (1 sample). Food samples purchased in Korea were produced using imported soybeans from America, Canada, India and Australia, and there were no samples labeled as GMO in the 23 samples.

### 2.2. DNA Extraction

Genomic DNA was extracted from the CRMs of the soybeans and processed food samples (approximately 1 g) using the Fast Genomic DNA Extraction Kit (Pinucle, Gyeonggi, Korea) according to the manufacturer’s protocol. Some processed food samples (such as tofu and soymilk) were dehydrated using a centrifuge (10 min at 13,000 rpm), and the snack sample was ground using liquid nitrogen. Each DNA sample was quantified using the Quant-iT dsDNA HS Assay kit and a Qubit Fluorometer (Invitrogen, Carlsbad, CA, USA). The DNA purity was measured with a NanoDrop One UV-Vis Spectrophotometer (Applied Biosystems, Foster City, CA, USA). Extracted DNA was diluted to 20 ng/µL and stored at −20 °C until use.

### 2.3. qPCR Array

The genetic elements of seventeen GM soybean events were gathered (Table 1) to select the screening assays. A total of eight screening assays were chosen to maximize the number of GM soybean events detected. The RbcS4 promoter (for MON87701 and MON87751), the nos terminator (for RRS, FG72 and SYHT0H2), the E9 terminator (for MON89788, MON87705, MON87708 and MON87769), the pat gene (for DAS-44406-6, DAS-68416-4, DAS-81419-2, SYHT0H2, A2704-12 and A5547-127), and three GM soybean event-specific methods (for CV127, DP305423-1 and DP356043-5) were selected as screening assays (including the lectin gene). One plate can test duplicates of five samples. The qPCR kit was prepared for the monitoring test using this study. The 15 µL of a mixture containing master mix, primers and probe for each target was pre-dispensed into an 8-strip tube and stored at −20 °C until use.

### 2.4. Target Specific Primers and Probes

The target-specific primer sets and probes for P-RbcS4, T-E9, and pat were designed using the Primer Designer program, version 3.0 (Scientific and Educational Software, Durham, NC, USA). The primer sets and probes for the other events were obtained from the Korean Food Code [23]. All of the primer sets and probes were synthesized by Bionics (Seoul, Korea). The sequences and references of primers and probes are shown in Table 2.

### 2.5. PCR Condition

The 7500 Real-Time PCR System (Applied Biosystems, Foster City, CA, USA) was used for developing soybean-specific screening methods using TaqMan probes. The screening assay was performed in a final volume of 25 µL with 1 × TaqMan Universal Master Mix II, with UNG (Applied Biosystems), 500 nM of each primer, 200 nM of the probe and 50 ng of genomic DNA. The real-time PCR conditions were as follows: one cycle of 2 min at 50 °C, 10 min at 95 °C, 45 cycles of 30 sec at 95 °C and 1 min at 60 °C. The data were analyzed with the 7500 Real-time PCR System software version 2.3 (Applied Biosystems).

### 2.6. Estimation of the Limit of Detection (LOD)

MON87751, RRS, MON89788, DAS-44406-6, CV127, DP305423-1 and DP356043-5 were selected for the sensitivity tests of P-RbcS4, T-nos, T-E9, pat, CV127, DP305423-1 and DP356043-5, respectively. The DNA mixture was prepared with different GM % (v/v) using non-GMO soybean DNA. Each GM soybean event’s DNA (20 ng/µL) was diluted using non-GMO soybean DNA (20 ng/µL) to three points (0.1%, 0.05% and 0.01% GM ratio). The LOD test was performed in triplicate three times with three different experimenters.

## 3. Results

### 3.1. Development of the Soybean-Specific Screening qPCR Array

A total of 16 promoters, 15 terminators and 21 genes were used for the development of 17 GM soybean events (Table 1). Based on this data, the screening assays were selected through the following steps: First, genetic elements used in only one GM soybean event or derived from the soybean itself were excluded from the screening targets. Second, an event-specific assay was chosen in case there was no screening target. Last, the redundancy of genetic elements was considered over their frequency of use. As a result, three genetic elements and three event-specific methods were added to the GMO analysis method announced on the Korean Food Code instead of P-35S. Finally, eight screening assays were selected for lectin, T-nos, P-RbcS4, T-E9, pat, CV127, DP305423-1 and DP356043-5.

The qPCR array’s flow chart is simpler than that of the universal screening method, which means that the number of additional events to test was lower than that of the universal screening method (Figure 1). For example, a sample with a negative signal at P-35S and T-nos needed additional analysis to determine that it was non-GMO. Twelve GM soybean events that do not have P-35S or T-nos as genetic elements, making it impossible to determine the presence/absence of GMO when those 12 GM events are included in the sample. However, the user can judge that a sample is non-GMO after seven screening assays (except for lectin, the endogenous gene of the soybean) of the qPCR array test negative. By using this screening step, users can save their time and cost of analysis by lowering the number of possible events, which is expected to be mixed into samples.

Compared to the previous report that described soybean-specific screening methods using multi-plex PCR targeting nine screening assays [4], the number of screening assays in this method is decreased to eight; meanwhile, the coverage of both tests is equal. The user can also conduct analysis easily because the qPCR array doesn’t demand the use of complex skills (such as multi-plex PCR).

Each target of the qPCR array dispenses into one well in 8-strip each, and the one strip becomes one set to analyze a sample. Thus, assuming a user tests a sample in duplicates, five samples (plus a positive and negative control) can be tested at once. This shows that the qPCR array can analyze the largest number of samples at once, compared to other studies [14,16,20].

### 3.2. The Specificity and Sensitivity of Screening Assays

Primers and probes for three screening assays (the RbcS4 promoter, the E9 terminator and the pat gene) were developed and successfully amplified. The amplicon size of the targets was under 200 bp for analyzing processed foods [24,25,26]. The specificity of the screening assays was evaluated and confirmed using 17 GM soybean events approved in Korea. Each screening target’s positive signals were observed only in events having each target as a genetic element (Table 3).

To determine sensitivity, three different GM ratios (0.1%, 0.05% and 0.01%) were tested as a template. The sensitivity of eight screening assays was determined to the point at which at least 95% of the total replicated reactions showed a positive signal [24]. The LODs of each screening assay (except that of DP356043-5) contained 0.01% GM content, corresponding to four copies of the soybean genome in 50 ng of genomic DNA (Table 4). This result indicated that this screening assay satisfied the labeling requirements of GMOs for Korea and the European Union (EU) [27,28]. This confirmed that the qPCR array could be used for monitoring and inspecting imported food.

### 3.3. The Applicability Test Using Processed Foods

To test a qPCR array’s applicability, the soybean-specific screening kit in which the primer/probe set and PCR mixtures are pre-dispensed was developed. Using the qPCR kit, GMO screening analysis can be prepared, with the only additional step being the addition of a template DNA into each well. Therefore, users can save at least half an hour of time preparing the real-time PCR. Twenty-three processed foods containing soybeans were tested using the qPCR array kit. The samples were obtained from domestic markets in Korea, Japan and Thailand; they contained soybeans as ingredients but no other GM crops. The processed foods were divided into five groups according to the following product types: soybean powder, dried cereal, soy milk, tofu and snack. Samples were analyzed in duplicates for accurate detection. Samples, where amplification was observed in even one well within 40 cycles, were determined to be positive in line with the Korean Food Code. Positive signals were observed in 13 of the 23 samples (Table 5), and 9 of these were tofu. Screening targets were detected in 13 samples and contained T-E9, T-nos, and pat. Base on the flow chart (Figure 1), GM event-specific PCR was performed in the second step to identify the GM events. MON89788, RRS and A2704-12 were identified in these samples (data not shown), and the result indicated that the prevalence of GM soybean events is the same as in the EU [22]. In addition, we performed PCR with processed foods to verify the results obtained with the proposed assay using the universal screening method (Appendix A). There is no difference between the results of the qPCR array and those of the universal screening method. Furthermore, it was confirmed that the qPCR array is more effective than the universal screening method because the number of expected events is lower.

Only five 96-well plates were used to screen the 23 processed foods, about half the amount used in other studies. This means that the user saves time and money using the qPCR array when screening for GM soybean events in food. In addition, the results showed that the qPCR array developed in this study was successful in detecting GM soybean events in food.

### 3.4. The Prediction System

When a positive signal was observed at the screening step, the next step was to assess its identity and authorization status. Since the number of event-specific tests to be examined depends on the screening results, it is necessary to understand the genetic information of GM events accurately. However, such information is difficult to identify because it is scattered across numerous documents or websites, and is different from Korea’s approval status. The flow chart (Figure 1) reflects the approval status within Korea, making it easier to find Korean-associated GM events. Meanwhile, the prediction system targets 24 GM soybean single events, listed on the International Service for the Acquisition of Agri-biotech Applications (ISAAA) database, approved by at least one country.

A prediction system was developed to reduce the time and effort required to analyze the screening results. This system does not need specific programs or access to a website and is in a simple-to-use Excel format (Figure 2). The file consists of the following two parts: one is the (Results) section (where the screening results are entered) and the other contains the genetic information of GM soybean events. A user inputs a 1 next to the positive screening target results in the (Results) section. Then, the number of expected GM events will be shown in the cell named (No. of events). Finally, users can confirm the expected GM events (labeled (Events) in the genetic information section). To identify specific GM soybean events, the user only needs to examine the prediction system’s event output. Therefore, a person from any background can analyze the presence of GM soybean events using this qPCR array. It is also worth noting that the prediction system was built using a program that is common throughout the world. Furthermore, the file can be modified to reflect different regulatory situations and will be provided via personal contact.

## 4. Conclusions

A systemic qPCR array was developed for the simple and rapid detection of GM soybeans. Eight screening assays were adopted and successfully tested using 17 GM soybean events and 23 processed foods. Compared to other studies, the number of screening targets were reduced, and flexibility and simplicity increased. Since each assay can be carried out independently, screening targets can be changed according to the user’s needs, and users can make their own qPCR array kit for rapid analysis. The prediction system can also be modified to suit the regulation system of any country. Because the GM soybean qPCR array focused on GM soybeans, this study has limitations in analyzing food samples containing a complex matrix. However, we expect that this screening system might be a useful tool for detecting GM soybeans in quarantine and monitoring systems due to the high consumption and import of soybeans around the world.

## Figures and Tables

**Figure 1 foods-10-00610-f001:**
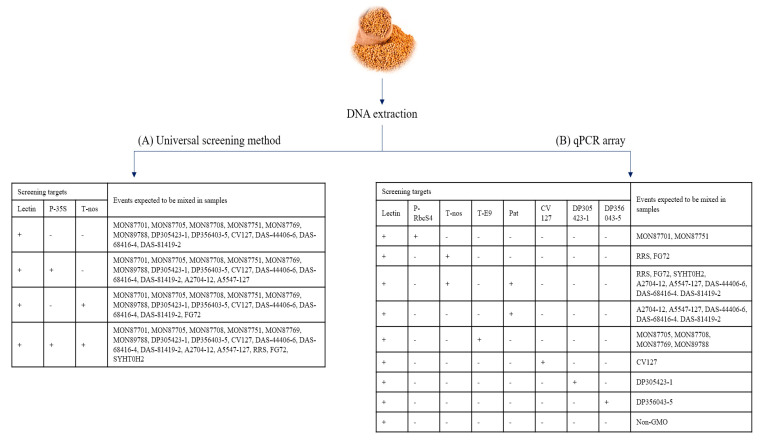
The comparison of the workflows of the two GM soybean screening methods ((+) and (−) represent positive and negative results in screening analysis, respectively). (**A**) The workflow of the universal screening method: This method analyzes three screening targets, including the endogenous gene of the soybean. Even though negative signals at P-35S and T-nos were observed, an additional analyzing step is needed to determine the presence/absence of GMO. (**B**) The workflow of the qPCR array developed in this study: the sample, confirmed all negative signals except for that of lectin at the screening analysis, can be judged to be a non-GM soybean.

**Figure 2 foods-10-00610-f002:**
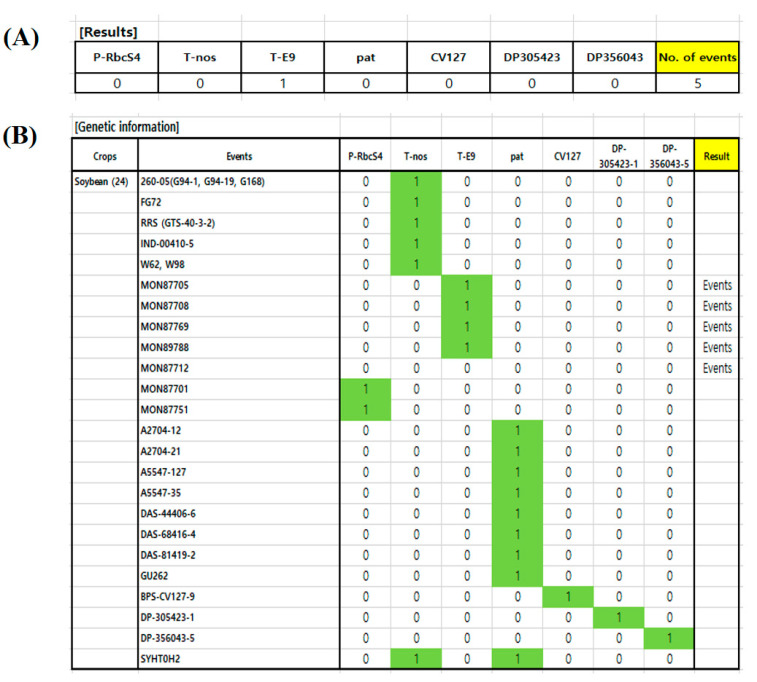
The prediction system combined with the qPCR array. (**A**), The result section; (**B**), the genetic information section. 1: Input 1 at screening target observed positive signal in the results section. 2: Check the (No. of events). 3: Confirm the expected GM events and plan the next step to identify the specific GM events.

**Table 1 foods-10-00610-t001:** Genetic elements of 17 GM soybean events.

Soybean Events	Genetic Elements		
	Promoter	Terminator	Gene
RRS	35S	Nos	Cp4 epsps
A2704-12	35S	35S	pat, bla
A5547-127	35S	35S	pat, bla
DAS-81419-2	CsVMV, Ubi10_At	tr7, ORF23	pat, cry1F, cry1Ac (synpro)
DAS-68416-4	CsVMV, Ubi10_At	tr7, ORF23	pat, aad-12
DAS-44406-6	CsVMV, Ubi10_At, h4a748	tr7, ORF23, h4a748	pat, aad-12, 2mepsps
MON87769	7Sα′, 7Sα	E9, tml	Δ6D_PJ, Fad3_NC
MON87705	FMV/TSF1, 7Sα′	E9, H6	Cp4 epsps, fad2-1A, FATB1-A
MON89788	FMV/TSF1	E9	Cp4 epsps
MON87701	RbcS4	7Sα’	cry1Ac
MON87751	RbcS4, act2	Mt, Pt1	cry1A.105, cry2Ab2
MON87708	PCSV	E9	dmo
DP305423-1	SAMS, Kti3	ALS_Gm, Kti3	hra_Gm, fad2-1_Gm
DP356043-5	SAMS, SCP1	PinII, ALS_GM	gat4601, hra_Gm
SYHT0H2	35S, CMP, SMP	Nos	pat, avhppd
CV127	AHASL	AHASL	csr1-2
FG72	h4a748	nos, h4a748	2mepsps, hppdPfW336

**Table 2 foods-10-00610-t002:** The sequences of primers and probes.

Targets	Primers/Probe	Sequences (5′→3′)	References
lectin	GM1-F	CCA GCT TCG CCG CTT CCT TC	[23]
	GM1-R	GAA GGC AAG CCC ATC TGC AAG CC	
	GM1-P	FAM-CTT CAC CTT CTA TGC CCC TGA CAC-TAMRA	
T-nos	NOS ter 2-5′	GTC TTG CGA TG ATTA TCA TAT AAT TTC TG	[23]
	NOS ter 2-3′	CGC TAT ATT TTG TTT TCT ATC GCG T	
	NOS-Taq	FAM-AGA TGG GTT TTT ATG ATT AGA GTC CCG CAA-TAMRA	
T-E9	T-E9_MF	CGA CAA CGT TCG TCA AGT TC	This study
	T-E9_MR	CCC AAT GCC ATA ATA CTC GAA C	
	T-E9_NP	FAM-AAT GCA TCA GTT TCA TTG CG-TAMRA	
P-RbcS4	P-RbcS4-F	AAG CAC CAC TCC ACC ATC AC	This study
	P-RbcS4-R	AGG TGT TGA GAC CCT TAT CG	
	P-RbcS4-P	FAM-ACG TGG CAT TAT TCC AGC GG-TAMRA	
pat	Pat-F	CGA TCC ATC TGT TAG GTT GC	[6]
	Pat-R	CCT TGG AGG AGC TGG CAA CT	
	Pat-P	FAM-ATA CAA GCA TGG TGG ATG GC-TAMRA	This study
CV127	SE-127-f4	AAC AGA AGT TTC CGT TGA GCT TTA AGA C	[23]
	SE-127-r2	CAT TCG TAG CTC GGA TCG TGT AC	
	SE-127-p3	FAM-TTT GGG GAA GCT GTC CCA TGC CC-TAMRA	
DP 305423-1	DP305423-f1	CGT GTT CTC TTT TTG GCT AGC	[23]
	DP305423-r5	GTG ACC AAT GAA TAC ATA ACA CAA ACT A	
	DP305423-1p	FAM-TGA CAC AAA TGA TTT TCA TAC AAA AGT CGA GA-TAMRA	
DP 356043-5	DP356-f1	GTC GAA TAG GCT AGG TTT ACG AAA AA	[23]
	DP356-r1	TTT GAT ATT CTT GGA GTA GAC GAG AGT GT	
	DP356-p	FAM-CTC TAG AGA TCC GTC AAC ATG GTG GAG CAC-TAMRA	

**Table 3 foods-10-00610-t003:** The results of the specificity test.

Soybean Events		Ct Value ± SD *						
	lectin	P-RbcS4	T-nos	T-E9	pat	CV127	DP305423-1	DP356043-5
RRS	25.42 ± 0.18	- **	28.37 ± 0.06	-	-	-	-	-
A2704-12	24.89 ± 0.02	-	-	-	20.85 ± 0.05	-	-	-
A5547-127	25.78 ± 0.14	-	-	-	22.99 ± 0.01	-	-	-
DAS-81419-2	26.43 ± 0.00	-	-	-	23.60 ± 0.00	-	-	-
DAS-68416-4	26.63 ± 0.01	-	-	-	24.64 ± 0.03	-	-	-
DAS-44406-6	25.31 ± 0.13	-	-	-	24.10 ± 0.00	-	-	-
MON87769	25.59 ± 0.05	-	-	27.27 ± 0.10	-	-	-	-
MON87705	25.61 ± 0.02	-	-	26.21 ± 0.08	-	-	-	-
MON89788	26.07 ± 0.18	-	-	26.22 ± 0.07	-	-	-	-
MON87701	25.25 ± 0.05	26.33 ± 0.05	-	-	-	-	-	-
MON87751	25.66 ± 0.02	25.95 ± 0.01	-	-	-	-	-	-
MON87708	25.32 ± 0.17	-	-	26.71 ± 0.09	-	-	-	-
DP305423-1	24.94 ± 0.10	-	-	-	-	-	25.43 ± 0.08	-
DP356043-5	25.67 ± 0.14	-	-	-	-	-	-	26.98 ± 0.01
SYHT0H2	26.40 ± 0.01	-	24.91 ± 0.02	-	20.54 ± 0.01	-	-	-
CV127	25.63 ± 0.05	-	-	-	-	25.95 ± 0.02	-	-
FG72	26.64 ± 0.02	-	26.84 ± 0.01	-	-	-	-	-

* Average Ct value ± SD obtained from duplicate reactions. ** (-) means not detected.

**Table 4 foods-10-00610-t004:** The results of the sensitivity test.

Target	Template	Positive Signal		
		Experimenter A	Experimenter B	Experimenter C
P-RbcS4	0.1%	9/9	9/9	9/9
	0.05%	9/9	9/9	9/9
	0.01%	9/9	9/9	9/9
T-nos	0.1%	9/9	9/9	9/9
	0.05%	9/9	9/9	9/9
	0.01%	9/9	9/9	9/9
T-E9	0.1%	9/9	9/9	9/9
	0.05%	9/9	9/9	9/9
	0.01%	9/9	9/9	9/9
pat	0.1%	9/9	9/9	9/9
	0.05%	9/9	9/9	9/9
	0.01%	9/9	9/9	9/9
CV127	0.1%	9/9	9/9	9/9
	0.05%	9/9	9/9	9/9
	0.01%	9/9	9/9	9/9
DP305423-1	0.1%	9/9	9/9	9/9
	0.05%	9/9	9/9	9/9
	0.01%	8/9	9/9	9/9
DP356043-5	0.1%	9/9	9/9	9/9
	0.05%	9/9	9/9	9/9
	0.01%	7/9	9/9	9/9

**Table 5 foods-10-00610-t005:** The results of the monitoring test using processed foods.

Product Type	Sample No.	Results								Events Expected to Be Mixed in Samples
		lectin	P-RbcS4	T-nos	T-E9	Pat	CV127	DP305 423-1	DP356 043-5	
Soybean Powder	1	+	-	-	-	-	-	-	-	-
	2	+	-	-	+	-	-	-	-	MON89788, MON87705, MON87708, MON87769
	3	+	-	-	-	-	-	-	-	-
Dried Cereal	4	+	-	-		-	-	-	-	-
	5	+	-	-		-	-	-	-	-
	6	+	-	-		-	-	-	-	-
	7	+	-	-	+	-	-	-	-	MON89788, MON87705, MON87708, MON87769
	8	+	-	-		-	-	-	-	-
	9	+	-	-		-	-	-	-	-
	10	+	-	-		-	-	-	-	-
	11	+	-	-	+	-	-	-	-	MON89788, MON87705, MON87708, MON87769
Soy Milk	12	+	-	+	+	+	-	-	-	RRS, FG72, MON89788, MON87705, MON87708, MON87769, A2704-12, A5547-127, DAS-44406-6, DAS-68416-4, DAS-81419-2, SYHT0H2
Tofu	13	+	-	+	+	+	-	-	-	RRS, FG72, MON89788, MON87705, MON87708, MON87769, A2704-12, A5547-127, DAS-44406-6, DAS-68416-4, DAS-81419-2, SYHT0H2
	14	+	-	-	-	-	-	-	-	-
	15	+	-	+	+	+	-	-	-	RRS, FG72, MON89788, MON87705, MON87708, MON87769, A2704-12, A5547-127, DAS-44406-6, DAS-68416-4, DAS-81419-2, SYHT0H2
	16	+	-	+	+	+	-	-	-	RRS, FG72, MON89788, MON87705, MON87708, MON87769, A2704-12, A5547-127, DAS-44406-6, DAS-68416-4, DAS-81419-2, SYHT0H2
	17	+	-	+	+	+	-	-	-	RRS, FG72, MON89788, MON87705, MON87708, MON87769, A2704-12, A5547-127, DAS-44406-6, DAS-68416-4, DAS-81419-2, SYHT0H2
	18	+	-	+	+	+	-	-	-	RRS, FG72, MON89788, MON87705, MON87708, MON87769, A2704-12, A5547-127, DAS-44406-6, DAS-68416-4, DAS-81419-2, SYHT0H2
	19	+	-	+	+	-	-	-	-	RRS, FG72, MON89788, MON87705, MON87708, MON87769
	20	+	-	+	+	+	-	-	-	RRS, FG72, MON89788, MON87705, MON87708, MON87769, A2704-12, A5547-127, DAS-44406-6, DAS-68416-4, DAS-81419-2, SYHT0H2
	21	+	-	+	+	-	-	-	-	RRS, FG72, MON89788, MON87705, MON87708, MON87769
	22	+	-	+	+	-	-	-	-	RRS, FG72, MON89788, MON87705, MON87708, MON87769
Snack	23	+	-	-	-	-	-	-	-	-

(+) and (-) represent positive and negative result in screening analysis, respectively.

## Data Availability

Not applicable.

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
