# Peer review of "Development of a Systematic qPCR Array for Screening GM Soybeans"

_foods, 2021, doi:10.3390/foods10030610_

Round 1

Reviewer 1 Report

The study describes the development of a qPCR array for screening soybeans to detect GMOs. The authors tested eight targets combined with a predictions system to identify GM soybeans.

These results show the possibility to use this assay for fast and sensitive detection of GM soybeans. The topic of this paper is interesting and could provide a useful assay for screening purposes, however, the manuscript needs major revisions, mostly related to the experimental data analysis, format and English form.

MATERIALS & METHODS: please describe why the authors chose the 17 listed events as reference materials

Paragraph 2.3: it would be clearer if the authors would list the genetic markers used in this assay, rather than indicating alternatively genes, events and genes again. Maybe authors could indicate the events in brackets next to the specific target gene.

RESULTS

Line 154 and throughout the text: most numbers of figures and tables in the text are wrong (they do not correspond to the correct fig/table)

The authors should include a regression analysis of the sensitivity experiments, maybe replacing figure 5.

The authors should test the commercial products included in the study also with the universal screening method in order to verify the results obtained with the proposed assay and confirm its suitability for screening purposes

Can the authors confirm positive signals by identyfing the GMOs containing the detected GM-elements.

MINOR POINTS:

Table 3: add the header in the first column

Line 113: the target P-RbcS4 is reported as  P-RbcS in the table 1

I suggest to have your manuscript revised by a native English speaker.

Author Response

We are very thankful to the reviewers for their comprehensive and thorough review. Our detailed response to the review comments are given below (in blue). We have addressed all of the concerns of the reviewers and hope that the revised manuscript is now suitable for publication.

Reviewer 1

The study describes the development of a qPCR array for screening soybeans to detect GMOs. The authors tested eight targets combined with a predictions system to identify GM soybeans.

These results show the possibility to use this assay for fast and sensitive detection of GM soybeans. The topic of this paper is interesting and could provide a useful assay for screening purposes, however, the manuscript needs major revisions, mostly related to the experimental data analysis, format and English form.

MATERIALS & METHODS: please describe why the authors chose the 17 listed events as reference materials

→ We chose the 17 listed GM soybean events approved as food in Korea and described in lines 15-17.

 Lines 15-17: Each assay’s specificity was tested and confirmed using seventeen GM soybean events that have been approved in Korea.

Paragraph 2.3: it would be clearer if the authors would list the genetic markers used in this assay, rather than indicating alternatively genes, events and genes again. Maybe authors could indicate the events in brackets next to the specific target gene.

→ As you recommended, we modified the sentence in lines 114-119 as follows:

Lines 114-119; The RbcS4 promoter (for MON87701 and MON87751), nos terminator (for RRS, FG72, and SYHT0H2), E9 terminator (for MON89788, MON87705, MON87708, and MON87769), pat gene (for DAS44406-6, DAS68416-4, DAS81419-2, SYHT0H2, A2704-12, and A5547-127), 3 GM soybean event-specific methods (for CV127, DP305423-1, and DP356043-5) were selected as screening assays including lectin gene.

RESULTS

Line 154 and throughout the text: most numbers of figures and tables in the text are wrong (they do not correspond to the correct fig/table)

→ As you recommended, we corrected most numbers of figures and tables in manuscript.

The authors should include a regression analysis of the sensitivity experiments, maybe replacing figure 5.

→ Thank you for your kind comments. This manuscript is about qualitative analysis of GM soybean. We suggest that it is sufficient to determine the sensitivity for this study by showing the number of positive signals as PCR results of the template % concentration. We deleted Ct values in Table 4 and do not include a regression analysis of the sensitivity experiments.

The authors should test the commercial products included in the study also with the universal screening method in order to verify the results obtained with the proposed assay and confirm its suitability for screening purposes

Can the authors confirm positive signals by identyfing the GMOs containing the detected GM-elements.

→ As you recommended, we performed PCR with processed foods to verify the results obtained with the proposed assay using universal screening method. The results were shown in supplementary data 1 and discussed the sentence in lines 233-237 as follows:

Lines 233-237; In addition, we performed PCR with processed foods to verify the results obtained with the proposed assay using universal screening method (Supplementary data 1). There is no different result between the qPCR array and the universal screening method. Furthermore, it was confirmed that the qPCR array is more effective better than the universal screening method because the number of expected events is lower.

MINOR POINTS:

Table 3: add the header in the first column

We added the header in the first column as you recommended.

Line 113: the target P-RbcS4 is reported as P-RbcS in the table 1

We corrected the target P-RbcS4 in the table 1 as you recommended.

I suggest to have your manuscript revised by a native English speaker.

This original manuscript was edited professionally by Enago Editing Service (https://www.enago.com; reference number: INQ-8156170820). However, we will edit this manuscript in another proofreading service after responding to comments from a third reviewer.

Reviewer 2 Report

  • Lines 20-21: In the abstract, it is stated: Fourteen of twenty-three samples included GM soybean as an ingredient. Table 5 shows that GM soy was detected in 13 samples tested. The results obtained by analysing 23 samples, i.e. the declaration of GM soy by the manufacturer in 14 samples and confirmation of the presence of GM soy in 13 samples (it was a subset of declared?), it would be appropriate to explain and discuss appropriately in the article.

  • Lines 26-28: Keywords are missing

  • Lines 99-107: How was homogenisation of processed food samples done? How many mg of sample was used for DNA extraction?

  • Lines 101 – 122: The authors of the article could consider merging parts A, B and C of Table 1 into one table; i.e. preserving the rows belonging to individual GM soybean events, but putting all the columns from parts A, B and C into one row. In my opinion, this would make the table clearer to readers.

  • Lines 123 – 124: I suggest deleting this image. These data (targets) are given in other pictures / tables and text of the article.

  • Line 143: I recommend adding a SW version.

  • Line 144: I would prefer the title: Estimation of the Limit of Detection (LOD)

  • Lines 144-150: How were DNA mixtures prepared? Please, describe.

  • Lines 144-150: Which percentage authors mean? Mass to mass, haploid genome equivalent (HGE) of another one?

There is not explained in the chapter 2: Materials and Methods, how was prepared pre-dispensed qPCR kit.

  • Line 165: delete: shown in Figure 1

  • Line 187: Figure 2, part A: In the section A of the Figure 2 I see the confusing information given in the column called "possible events". For example, DAS68416-4 / DAS81419-2 / CV127 are listed as possible events for all 4 combinations of results obtained by analysis of the three targets, but should only be in the last row (Le +, P-35S - and T-nos -). In the third line (Le +, P-35S - and T-nos +) only GM soy event FG72 should be listed etc..

  • Line 192: The sample shown all negative signals except lectin at the screening analysis can be judged to Non-GMO authorised soybean.

  • Line 200: To facilitate the reading (clarity) of the text, I would recommend to unify the order of GM events in Tables 1 and 3.

  • Table 3, 4 and 5: I would appreciate the addition of SD of observed Ct-values

  • Lines 210-211: What is authors explanation of results obtained in analysis of DP356043-5? I know that the article is about qualitative detection by qPCR, not quantification. Difference 4.4 cycle for parallel sample (without SD measurement listened) seems no very good.

  • Line 218: Sample from America? In previous text is written that the third country (beside Korea and Japan) was Thailand (line 96).

  • Line 224: Positive signals were observed in 14 of 23 samples (Table 5), and nine were tofu. There is only 13 positive samples listed in Table 5.

  • Line 234: Table 5. I am sorry I do not understand what the authors of the study mean by the information given in the column called "possible events", e.g. RRS is not pat positive etc.

  • Line 234: Table 5. I propose to add information to the table as to whether the presence of genetically modified soy was indicated by the manufacturer in the analysed products.

  • Lines 246-258: Developed simple-to-use excel format seems very useful. Will it be a part of publication, for example available as supplementary data? If yes, please, write link into this paragraph.

  • Lines 259-260: Values of Ct on Figure 3 are not in correspondence it results of specificity test written in Table 3.

  • Lines 261-264: Figure A: Why is referred No. of events 5, it in table 1 is only 4 GM soybean with T-E9 as well as in part B of figure 4?
